# Configuration Design and Verification of Shear Compliant Border in Space Membrane Structure

**DOI:** 10.3390/polym16070951

**Published:** 2024-03-30

**Authors:** Anbo Cao, Zhiquan Liu, Qiuhong Lin, Hui Qiu

**Affiliations:** Beijing Institute of Spacecraft System Engineering, Beijing 100094, China; anbocao9508@gmail.com (A.C.); qh_sofa@126.com (H.Q.)

**Keywords:** spacecraft, membrane, kapton, shear compliant border, stress uniformity

## Abstract

To solve the non-uniformity of stress in space membrane structure and the lack of shear compliant border configuration design method, shear compliant borders are designed, optimized, and verified in terms of configuration. Firstly, an orthotropic model of the borders is built by combining Hill and Christensen-Lo composite material models. Secondly, a finite element form-finding method is put forward by establishing rectangular and cylindrical coordinates in different areas. The configuration of borders is obtained and the influence of the borders on the edge of the membrane is 0.23%, which means that the borders are compatible with the existing tensegrity systems, especially the tensioning components and the cable sleeves. Thirdly, simulation verifies that borders can cut the spread of shear stress and improve the stress uniformity in membrane structure. The maximum stress in the membrane effective area is decreased by 35.6% and the stress uniformity is improved by 30.5%. Finally, a membrane extension experiment is committed to compare the flatness of membrane surface under shear stress with and without shear compliant borders. The borders decrease the increment speed of flatness by 58.1%, which verifies the amelioration of stress uniformity. The shear compliant border configuration design method provides a reference for space membrane structure stress-uniform design.

## 1. Introduction

Kapton is a high-performance polymer material synthesized by subjecting high-temperature treated polyamic acid solution to dehydration cyclization. It has excellent thermal stability and high temperature resistance, as well as good mechanical properties. Kapton is widely used in space membrane structures such as sunshades [1], antennas [2], and solar sails [3]. To meet the stiffness requirements of the membrane structure, tensile forces need to be applied through a tensioning system. However, the presence of seams and coatings within the membrane structure introduces non-uniform and discontinuous mechanical characteristics [4,5,6]. These features alter the stress distribution within the membrane under the traction of tensioning system, reducing stress uniformity [7]. Non-uniform stress can easily lead to membrane wrinkles, adversely affecting the performance of spacecraft products [8], and localized stress concentrations can exacerbate creep and shorten the lifespan of the membrane structure. Scholars have actively explored solutions to improve stress uniformity within membrane structures under tensegrity systems. Since 2002, Shear Compliant Borders (SCB) have emerged [9,10], initially applied in the American SRS (Scalable Readout System) solar sails. SCB is a method of locally pressurizing the membrane onto a mold with strip-like protrusions through vacuum or similar means, and then heating the adhered membrane and mold together to induce thermoplastic deformation in the membrane, resulting in an orthotropic structure. Its appearance is similar to the waistband of elastic pants [9]. SCB can intercept shear stress transmission between membrane regions, optimize stress distribution within the membrane structure, and improve stress uniformity. The configuration optimization design of SCB within space membrane structures is crucial as it directly affects or determines the application effect of SCB. In the configuration optimization design of SCB, an appropriate mechanical model needs to be established and incorporated into an algorithm for calculation. Conducting optimization design and verification of SCB within space membrane structures is of great significance for improving membrane structure performance and extending its lifespan.

In 2002, to address the issue of wrinkles in space membrane structures caused by shear stress after tensioning, the American SRS Technology Company, led by Talley et al., vacuum-adhered the edge region of the membrane to a mold with strip-like protrusions. Subsequently, the adhered membrane and mold were heated together using a hot air gun, resulting in the emergence of numerous narrow, relaxed plastic deformation strips arranged perpendicular to the edge direction of the membrane, thereby creating the SCB structure [10]. The width of the SCB region was a constant value parallel to the membrane edge, with the internal strips of the region parallel to each other and evenly spaced. Experimental results indicated that when the membrane structure was subjected to shear stress applied by the tensioning system, the shear strain in the membrane area on both sides of the SCB significantly decreased compared to when there was no SCB, demonstrating the role of SCB in intercepting shear stress transmission and suppressing wrinkles in the inner membrane area of the SCB. However, the article [10] did not quantitatively study the relationship between SCB design parameters such as width and strip ratio (the ratio of the area undergoing deformation to the entire heated area) and wrinkle suppression ability. In 2003, Leifer et al. from the University of Kentucky provided insights into the influence of SCB width on membrane wrinkles through simulation and experimentation [11]. In the simulation model and test specimens, SCB was placed adjacent to the edge of the membrane structure. In actual SCB preparation, however, a certain distance should be left between SCB and the membrane edge due to the vacuum equipment’s space requirements. This distance caused the unformed membrane within it to undergo strain under shear force, resulting in an overestimation of the measured shear strain in the article [11]. In 2007, Leifer treated SCB as a composite material consisting of relaxed and flat membranes arranged in parallel, proposing a relatively simplified method for calculating the Young’s modulus and Poisson’s ratio of SCB compared to the article [11], demonstrating that SCB could be equivalent to an orthotropic material [9]. However, Leifer’s model assumed that the deformed region of SCB was anisotropic, rendering the model relatively complex. In 2010, Fellini et al. from Northrop Grumman Corporation conducted a more detailed modeling of SCB, reproducing the phenomenon of the relaxation region being flattened as shear stress increased in the simulation model and proposing an analytical model to calculate SCB stiffness based on some design parameters of SCB [12]. However, Fellini’s model did not provide a method for calculating SCB’s Poisson’s ratio, limiting the model’s application scope. All the aforementioned studies assumed that SCB was a region of equal width parallel to the membrane edge, with all strips within SCB remaining parallel to each other, without optimizing the configuration of SCB. The membrane sunshield of the James Webb Space Telescope applied the research results from literature [12] to prepare SCB inside the Kapton membrane to suppress wrinkles [13]. The configuration of SCB within the membrane plane consisted of multiple segments of approximate arcs, with the strips intersecting at certain angles. However, the article [13] did not provide a design method for SCB configuration within the membrane structure, and important design parameters such as arc radius and strip direction could not be obtained from public literature. To obtain these parameters, a form-finding process is necessary. There are already some studies on the form-finding analysis of anisotropic material. Le Meitour put forward a form-finding approach based on dynamic relaxation method [14]. Le Meitour’s method could be used to simulate the inflation of a zero pressure balloon made of anisotropic material. Pinto et al. proposed a form-finding method based on finite element analysis and used their method to calculate the shape of an anisotropic thin shell under bending forces [15]. The form-finding of membrane structures with SCB requires a method that can be applied to a structure partially anisotropic and the rest isotropic, which is not found in public literature.

To solve the stress non-uniformity within space membrane structures and to fill the gap in the lack of SCB configuration design methods, this paper proposes an SCB configuration design method and verifies the effect of SCB in intercepting shear stress transfer and improving stress uniformity through simulation and experiments. Firstly, SCB is equivalently modeled as an orthotropic composite material composed of two isotropic materials, and a mechanical model of SCB is established using analytical methods. Secondly, a finite element form-finding method containing anisotropic material is proposed to obtain the configuration of SCB within the membrane structure. Then, the effect of SCB on intercepting shear stress transmission and optimizing stress uniformity is verified by simulation and experiment. The SCB configuration design method proposed in this paper provides an effective and feasible solution for stress uniformity design in space membrane structures.

## 2. Establishment of the Shear Compliant Border Mechanical Model

The James Webb Space Telescope utilizes SCB in its Kapton membrane sunshield, with the relative positioning of the SCB to the membrane structure depicted in Figure 1.

The inner side of the membrane structure is the effective area of the membrane, responsible for functions such as emitting electromagnetic waves, and it should be kept as smooth as possible with sufficient area to fulfill these functions. Therefore, SCB must be positioned close to the edge of the membrane structure. The edges of the membrane structure are tensioned using cables and membrane sleeves. During the preparation of SCB, the membrane needs to be vacuum-adhered to the mold, requiring a certain distance between the cables and the SCB. A schematic diagram of the prepared SCB is shown in Figure 2. The area between the dotted lines is the SCB.

During the preparation of SCB, the thermoplastic membrane and the mold with strip-like protrusions are first fitted together through evacuation. The width of protrusions, which is also the width of SCB, is *L*. The membrane undergoes elastic deformation at the protrusions of the mold. Then, the membrane is heated, causing the elastic deformation at the protrusions to transform into plastic deformation. The Young’s modulus and Poisson’s ratio of the deformed Kapton membrane are denoted as *E*_h_ and *ν*_h_ respectively, with the subscript h indicating heating. The mechanical properties of the undeformed membrane area remain unchanged after returning to room temperature. The Young’s modulus and Poisson’s ratio are the same as those of the Kapton membrane outside the SCB, denoted as *E*_m_ and *ν*_m_ respectively, with the subscript m indicating membrane. In this paper, both the deformed and undeformed membranes are considered to be isotropic materials. According to Hooke’s law, the shear moduli of the two materials are given by Equations (1) and (2), respectively [16]:(1)$$G_{h}=\frac{E_{h}}{2\left(1+\nu_{h}\right)}$$
(2)$$G_{m}=\frac{E_{m}}{2\left(1+\nu_{m}\right)}$$

The widths of the deformed and undeformed membranes in SCB are denoted as *l*_h_ and *l*_m_, respectively, while the thickness of the membrane remains constant at *t*. To simplify the calculation process, use *V*_h_ and *V*_m_ to represent the width ratios of the deformed and undeformed membranes, respectively. The calculation method is as follows:(3)$$V_{h}=\frac{l_{h}}{l_{h}+l_{m}}$$
(4)$$V_{m}=\frac{l_{m}}{l_{h}+l_{m}}$$

Since both the deformed and undeformed membranes have a length equal to the width of SCB, denoted as *L*, *V*_h_ and *V*_m_ are also the area ratios of the deformed and undeformed membranes, respectively. There is always *V*_h_ + *V*_m_ = 1.

SCB can be equivalent to a composite material consisting of two parallel distributed materials: the deformed membrane and the undeformed membrane. This composite material is an orthotropic material [16,17]. Denoting SCB with the subscript c (for “compliant”), the calculation of SCB’s Young’s modulus *E*_c_, Poisson’s ratio *ν*_c_, and shear modulus *G*_c_ in the coordinate system shown in Figure 2 is as follows.

SCB is a composite material with a series structure in the *y* and *z* directions. According to the assumption of equal strain:(5)$$E_{yc}=E_{h}V_{h}+E_{m}V_{m}$$
(6)$$\nu_{yxc}=\nu_{h}V_{h}+\nu_{m}V_{m}$$

SCB is a composite material with a parallel structure in the *x* direction, therefore:(7)$$E_{xc}=E_{zc}=\frac{E_{h}E_{m}}{E_{h}V_{m}+E_{m}V_{h}}$$
(8)$$\nu_{yzc}=\nu_{xyc}$$

According to the Hill composite material model [18], the shear modulus of SCB in the *xy* plane is given by:(9)$$G_{xyc}=\frac{G_{h}\left(G_{m}V_{h}+2G_{m}V_{m}+G_{h}V_{h}\right)}{G_{m}V_{h}+2G_{h}V_{m}+G_{h}V_{h}}$$

According to the Christensen-Lo composite material model [19], the shear modulus of SCB in the *xz* and *yz* planes is:(10)$$G_{xzc}=G_{yzc}=G_{h}\left[1+V_{m}{\left(\frac{G_{h}}{G_{m}-G_{h}}+\frac{V_{h}\left(k_{h}+\frac{7}{3}G_{h}\right)}{2k_{h}+\frac{8}{3}G_{h}}\right)}^{-1}\right]$$

The Poisson’s ratio is:(11)$$\nu_{xzc}=\frac{E_{xc}}{2G_{xzc}}-1$$
where *k*_h_ represents the bulk modulus of the deformed membrane. When the deformed membrane is an isotropic material, the calculation method for *k*_h_ is [16]:(12)$$k_{h}=\frac{E_{h}}{3\left(1-2\nu_{h}\right)}$$

For orthotropic materials:(13)$$\nu_{xyc}=\nu_{yxc}\left(\frac{E_{xc}}{E_{yc}}\right)$$

With this, a total of 9 parameters are obtained: *E_x_*_c_, *E_y_*_c_, *E_z_*_c_, *ν_xy_*_c_, *ν_xz_*_c_, *ν_yz_*_c_, *G_xy_*_c_, *G_zx_*_c_ and *G_yz_*_c_. These parameters can be used to formulate the constitutive equations of SCB.
(14)$$\left\{\begin{array}{c} \varepsilon_{x}\\ \varepsilon_{y}\\ \varepsilon_{z}\\ \gamma_{yz}\\ \gamma_{zx}\\ \gamma_{xy} \end{array}\right\}=\left[\begin{array}{cccccc} 1/E_{xc} & -\nu_{yxc}/E_{yc} & -\nu_{zxc}/E_{zc} & 0 & 0 & 0\\ -\nu_{xyc}/E_{xc} & 1/E_{yc} & -\nu_{zyc}/E_{zc} & 0 & 0 & 0\\ -\nu_{xzc}/E_{xc} & -\nu_{yzc}/E_{yc} & 1/E_{zc} & 0 & 0 & 0\\ 0 & 0 & 0 & 1/G_{yzc} & 0 & 0\\ 0 & 0 & 0 & 0 & 1/G_{zxc} & 0\\ 0 & 0 & 0 & 0 & 0 & 1/G_{xyc} \end{array}\right]\left\{\begin{array}{c} \sigma_{x}\\ \sigma_{y}\\ \sigma_{z}\\ \tau_{yz}\\ \tau_{zx}\\ \tau_{xy} \end{array}\right\}$$

According to Equation (14), the stress-strain relationship of SCB can be derived, thereby establishing the mechanical model of SCB. The main difference between our model and Leifer’s model [9] lies in the use of the Christensen-Lo composite material model to calculate the shear modulus of SCB in the *xz* and *yz* planes. This model requires the bulk modulus *k*_h_ of the membrane (Equation (12)). In Leifer’s model [9], the bulk modulus *k*_h_ is calculated through Equations (15)–(17):(15)$$G_{h}=\frac{E_{h}}{1+2\nu_{h}}$$
(16)$$K_{h}=\frac{E_{h}}{2\left(1-2\nu_{h}\right)\left(1+\nu_{h}\right)}$$
(17)$$k_{h}=K_{h}-\frac{G_{h}}{3}$$
where *K*_h_ is the compression modulus of the deformed membrane. In our model, the bulk modulus, *k*_h_, is calculated based on the assumption that the deformed membrane is an isotropic material, so that Equation (12) can be used to calculate *k*_h_. The assumption of Leifer’s model, especially the applicability of Equation (15), is not plainly illustrated in the article [9]. Using the design parameters in Table 1, MATLAB R2023a calculates *G_zx_*_c_ using our method in 6.37 × 10^−4^ s, resulting in *G_zx_*_c_ = 20.8066 MPa. Using the method from Leifer’s paper [9], the calculation of *G_zx_*_c_ takes 9.60 × 10^−4^ s, resulting in *G_zx_*_c_ = 20.3647 MPa. The difference is ((20.8066 − 20.3647)/20.3647) × 100% = 2.17%, and the calculation time is reduced by ((9.60 − 6.37)/9.60) × 100% = 33.6%.

## 3. Configuration Design of Shear Compliant Border Based on Finite Element Form-Finding

Using the spacecraft membrane structure as shown in Figure 3a as a case study, the configuration of SCB is computed.

The spacecraft membrane structure is square-shaped with an outer edge length of *L*_1_. The approximate square region on the inner side of the membrane structure is the effective area, with an edge length of *L*_2_. The SCB is located outside the effective area, with a width of *L*_3_. The distance from the SCB to the edge of the membrane structure is *L*_4_. There is always 2*L*_4_ + 2*L*_3_ + *L*_2_ = *L*_1_. Filleted corners are formed where the SCB connects to the effective area, with a radius of *R*_2_. The outer corners of the SCB are filleted with a radius of *R*_3_. There is always *R*_3_
*= R*_2_
*+ L*_3_. The values of *L*_1_ and *L*_2_ are determined by the performance requirements of the spacecraft and the manufacturing process of the SCB. In this study, *R*_2_, *R*_3_, *L*_3_, and other design parameters of the SCB, as well as the orientation and proportion of the strips in the SCB, are solved during configuration optimization design.

When establishing the model shown in Figure 3a in Abaqus 2023, it is necessary to specify the orientation of the material as SCB is an orthotropic material. Therefore, the SCB region in the figure is divided into four rectangular regions and four sector regions. The length direction of the rectangular regions corresponds to the *x* direction in Figure 2. The coordinate system of the left rectangular area, marked by horizontal lines, is illustrated in Figure 3b. For each sector region, a cylindrical coordinate system is established with the origin at the center of the membrane structure. The *ρ* direction in the cylindrical coordinate system is aligned with the *y* direction in Figure 2. The coordinate system of the lower right sector area, marked by diagonal lines, is shown in Figure 3b, as an example. The deformation strips in the SCB point towards the center of the membrane structure within each sector region. The thickness of all membranes is denoted by *t*. Point O represents the midpoint of the upper edge of the membrane structure.

The membrane structure edges are tensioned by cables with a radius of *r*, Young’s modulus of *E*_r_, and Poisson’s ratio of *ν*_r_, where the subscript r denotes rope.

The membrane used in simulation and experiment is made of Kapton^®^ 200HN, a material suitable for aerospace applications and manufactured by DuPont in Wilmington, DE, USA [20]. Its chemical structure is shown in Figure 3c. Referring to references [9,21,22,23], the design parameters of the example shown in Figure 3 are listed in Table 1.

During the form-finding process, it is necessary to apply prestress to the membrane structure [23]. As a planar aerospace structure (solar sail, sunshield or antenna), the entire structure should be tensioned and deform only in the *xy* plane. Uniform prestress is applied to the membrane within the membrane plane. As for the cable, it is considered to be one-dimensional truss element during form-finding analysis, so uniform prestress is applied along its axial direction. According to Hooke’s law:(18)$$\sigma_{\mathrm{ri}}=E_{r}\varepsilon_{\mathrm{ri}}$$
(19)$$\sigma_{\mathrm{mi}}=\frac{E_{m}}{1-\nu_{m}}\varepsilon_{\mathrm{mi}}$$
*σ*_ri_ and *σ*_mi_ represent the prestress applied to the cable and membrane, respectively, where the subscript i denotes the initial state. *ε*_ri_ and *ε*_mi_ represent the strains of the rope and membrane after applying the prestress. To prevent wrinkling of the membrane surface during tensioning and reduce friction between the rope and membrane, it is necessary to ensure that the strains of the cable and membrane match with each other. Referring to the article [21], taking *ε*_ri_ = *ε*_mi_ = 0.1%, the following prestress is obtained:(20)$$\sigma_{\mathrm{ri}}=20\;\mathrm{MPa}$$
(21)$$\sigma_{\mathrm{mi}}=3.34\;\mathrm{MPa}$$

The prestress *σ*_mi_ is applied to the effective area and the area between the SCB and the cables. According to the article [12], the SCB is not flattened when the shear displacement is 12% of the width of SCB (*L*_3_ in Figure 3a). When the shear displacement within the SCB is smaller than this value, the SCB will not cause wrinkles inside the membrane due to mismatched strains. Therefore, it is assumed that there is no initial prestress inside the SCB region in the initial state of the membrane structure. Hence, in the SCB region there is *σ*_ci_ = 0 MPa. If excessive shear stress is applied to the SCB, the corrugations will flatten and the SCB area will behave like a panel material tensioned diagonally, whose mechanical properties are yet to be studied. The shear displacement within SCB area is kept smaller than 12% of the width of SCB in all the simulations and experiments of this study.

In the finite element model, the membrane adopts M3D3 elements, while the cables adopt T3D2 elements. The membrane and cables are bound together, and fixed constraints are applied to the four corners of the membrane structure. Form-finding is a large deformation process, so the geometric nonlinearity switch is turned on during the analysis to constantly regenerate the stiffness matrix during the simulation. During the first form-finding iteration, the elastic modulus of the membrane, cables, and SCB is reduced by two orders of magnitude, and prestress and boundary conditions are applied. The obtained shape of the membrane structure is used as the result of the first iteration. Based on the result of the first iteration, the real elastic modulus of each part is reapplied, and the shape of the membrane structure is recalculated as the result of the second iteration. Keeping the real elastic modulus of each part, the shape of the membrane structure is continuously calculated. When the maximum displacement of the structure after calculation is less than 1 mm, the result is considered to have converged. Thus, the final iteration result is obtained, and the shapes of the membrane part and SCB are shown in Figure 4.

From Figure 4, it can be observed that the design parameters *R*_2_, *R*_3_, and *L*_3_ obtained after form-finding are not constant values. The configuration of SCB is an irregular shape composed of multiple approximate circular arcs spliced together. The direction of the SCB after form-finding is obtained from the direction of each element in the finite element software, as shown in Figure 5.

In Figure 5, the direction of the strips is marked by red arrows. The direction in the corner area of the SCB after form-finding differs significantly from that in the longer arc-edge area. The direction of the strips in the corner area is approximately parallel.

To investigate the impact of introducing SCB into the spacecraft membrane structure, the SCB area in Figure 3 is set to be the same material as the membrane, and the form-finding result without SCB is calculated. Furthermore, two other form-finding results are obtained by varying *L*_4_ = 20 mm in Figure 3 to 10 mm and 30 mm. A comparison of the membrane edge can be seen in Figure 6, with the positions on the *x* and *y* axes as shown in Figure 4. The central part of Figure 6a, circled with dashed box, is magnified in Figure 6b.

It can be observed that although the upper edges in form-finding results with different *L*_4_ misalign with each other, their central points are approximately at the same position. In contrast, the upper edge center of the membrane without SCB is 0.4 mm higher than any structure with SCB. Such a comparison shows clearly that the existence of SCB alters the membrane configuration.

Nevertheless, the maximum difference in the upper edge between the form-finding results with *L*_4_ = 20 mm and without SCB is 1.16 mm, which is (1.16/500) × 100% = 0.23% of the membrane structure’s size. As the manufacturing of aerospace membrane structure has a precision of ±1 mm, a difference of no larger than 1.16 mm is tolerable, which means that the existing tensioning components and cable sleeves can be directly used on a new membrane structure with SCB. It is also possible to prepare SCB on the already assembled membrane structures if the manufacturing technique allows. The comparison in Figure 6 demonstrates excellent compatibility between SCB and the existing membrane structures.

## 4. Simulation Verification of Shear Compliant Border’s Effect on Stress Uniformity Improvement

The primary source of shear stress experienced by the spacecraft’s membrane structure is introduced during the fabrication of the membrane sleeve. To simulate and verify the effect of SCB in intercepting shear stress transmission and improving stress uniformity, a lateral displacement of 5 mm to the right is applied to the midpoint O of the upper edge of the membrane structure in the example shown in Figure 3. This loading condition is similar to situations where shear stress is induced due to misalignment during the assembly of the sleeve.

For the membrane structure with and without SCB, the stress distribution in the membrane and the effective area after applying shear stress is illustrated in Figure 7.

Comparing the effective area in Figure 7b,d, it can be observed that when SCB is present, less shear stress is transmitted from point O to the effective area. The maximum stress in the effective area is reduced compared to the case without SCB by ((3.003 − 1.934)/3.003) × 100% = 35.6%. This indicates that SCB can effectively intercept the transmission of shear stress.

Taking the mesh nodes within the effective area in the finite element model as sampling points, the number of points is *n*. The von Mises stress of the *i*-th sampling point is $\sigma_{\mathrm{Mises}}^{i}$, the standard deviation of the von Mises stress of all sampling points is defined as the stress uniformity parameter *β* [24,25].
(22)$$\beta =\sqrt{\frac{1}{n-1}\sum_{i=1}^{n}{\left(\sigma_{Mises}^{i}-\bar{\sigma_{Mises}}\right)}^{2}}$$

A smaller *β* value indicates a more uniform stress distribution.

In the simulation results of Figure 7, there are 2256 mesh nodes in the effective area, therefore *n* = 2256. Substituting the results from Figure 7 into Equation (22), the stress uniformity of the effective areas with and without SCB are obtained as *β* = 3.48 MPa and *β* = 5.01 MPa, respectively. SCB reduces *β* by ((5.01 − 3.48)/5.01) × 100% = 30.5%, indicating a significant improvement in stress uniformity.

## 5. Experimental Validation of Stress Uniformity Improvement Effect of Shear Compliant Border

To compare the influence of the presence or absence of SCB on the flatness of the membrane under shear strain and to experimentally verify the improvement effect of SCB on stress uniformity, a membrane tension test is conducted. The experiment sample and setup are shown in Figure 8.

The experimental approach outlined in reference [11] is adopted in this study, substituting hollow membrane strips for SCB. The existing fabrication technology is combined with the design of SCB configuration. The width, density, and other design parameters of SCB are determined using the configuration calculation method proposed in this paper. The designed membrane test specimen is depicted in Figure 8b. Letters A-G are used to mark 8 points on the membrane. The boundaries of the specimen, labeled as ABCD, form a rectangle symmetrically along its centerline. The square region EFGH serves as the effective area of the membrane structure.

The design parameters of the example shown in Figure 8b are presented in Table 2.

The AB and CD edges of the membrane are fixed between plate A and plate B. The edges are positioned and fixed through a stepped structure. Plate A and plate B are connected by screws. Plate C is fixed with the frame through screws. Plate B and plate C are connected by bolts where the connection is an oblong hole, so that the combination of plate A, plate B and membrane edge AB or CD can move in the *x* direction while the tensile stress in the membrane remains unchanged within a small range in the *y* direction. A total of 16 sampling points are pasted in a square grid pattern in the EFGH area, and their spatial coordinates are obtained through photogrammetry. The photogrammetry device takes 30 photos of the whole experiment setup and calculates the coordinates of the 16 sampling points. The plane of the membrane is obtained by fitting these coordinates. The standard deviation of the distances of these 16 sampling points relative to the membrane plane is defined as the flatness (Root Mean Square, RMS) of the membrane’s effective area EFGH.

During the experiment, the frame containing edge CD is firstly moved along the *x*-direction to apply a displacement Δ*L* to the boundaries of the membrane test specimen. With the shear stress, the specimen will form a shape similar to a parallelogram, and edge CD will move along the positive direction of *y*-axis. At the same time, the frame ensures that the specimen is subjected to constant tensile stress and remains tensioned. Then, photogrammetric measurements are taken to determine the flatness of the effective area. Another membrane test specimen with no SCB but the same design parameters as shown in Figure 8b is subjected to the same displacement Δ*L*, and the flatness of the specimen is recorded. Five measurements are committed for each Δ*L*. The mean value and the square deviation of each measurement are acquired. The experimental results are shown in Figure 9.

For the membrane with SCB, the flatness is fitted as:(23)$$\mathrm{RMS}=0.13\Delta L+0.866,\;R^{2}=0.956$$
while for the membrane without SCB, the flatness is fitted as:(24)$$\mathrm{RMS}=0.31\Delta L+0.876,\;R^{2}=0.981$$

According to the fitted experimental results, for every 1 mm of displacement applied to the membrane edge, the average flatness of the specimen with SCB increases by 0.13 mm, while the average flatness of the specimen without SCB increases by 0.31 mm. This indicates that SCB reduces the rate of increase in flatness by ((0.31 − 0.13)/0.31) × 100% = 58.1%. SCB exhibits a significant inhibitory effect on the change of membrane flatness caused by shear stress, suggesting that SCB improves the stress uniformity of the membrane structure.

## 6. Conclusions

After conducting the optimization design and validation of SCB in space membrane structures, the following research conclusions have been obtained:(1)A shear compliant border configuration design method is proposed. The method is based on anisotropic finite element form-finding. Borders designed according to this method are proved to have the effect of intercepting shear stress transmission between membrane regions, optimizing stress distribution, and improving stress uniformity. The simulation shows a reduction of maximum stress by 35.6% within the effective area, and a decrease of 30.5% on stress uniformity. The experiment demonstrates a 58.1% reduction in the rate of increase in flatness.(2)The addition of shear compliant border to the membrane structure, whose configuration is designed by the method proposed in this research, has an impact of 0.23% on the shape of the membrane structure’s edge. This tolerable difference demonstrates excellent compatibility between SCB and the existing membrane structures. The ready-made tensioning components and cable sleeves can be used directly on a new membrane structure with SCB. SCB can also be prepared on the already assembled membrane structures as long as the manufacturing technique allows.

## Figures and Tables

**Figure 1 polymers-16-00951-f001:**
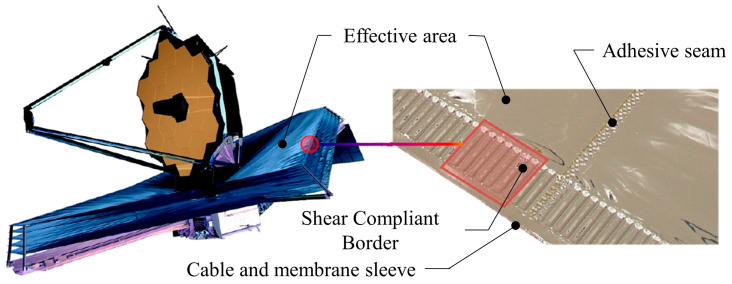
Membrane structure of JWST.

**Figure 2 polymers-16-00951-f002:**
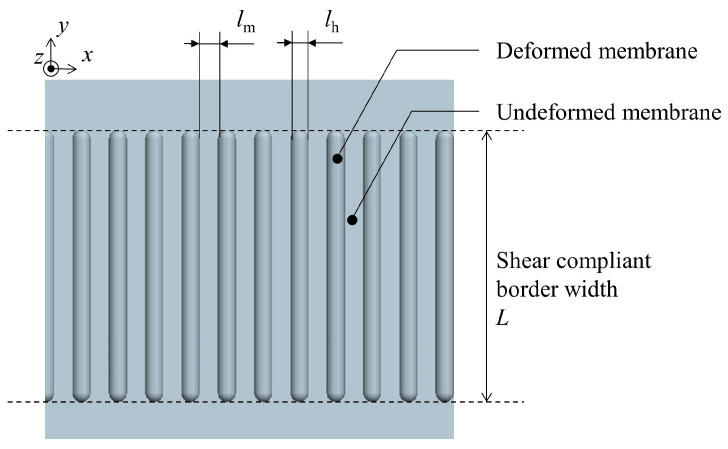
Diagram of shear compliant border.

**Figure 3 polymers-16-00951-f003:**
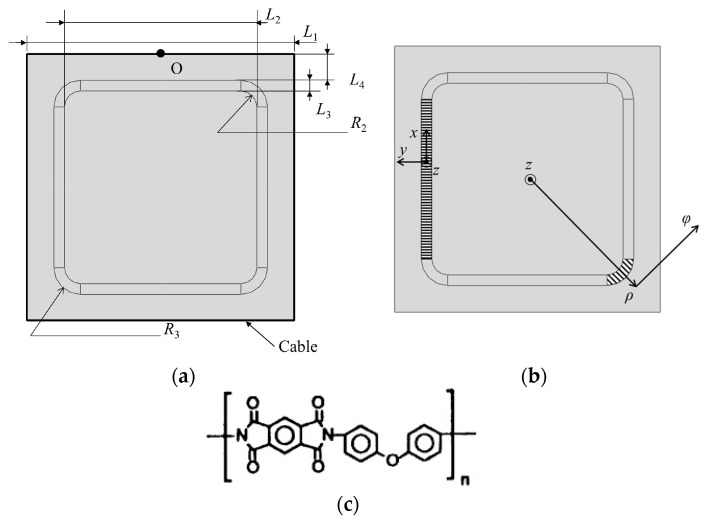
(**a**) Diagram of space membrane structure, (**b**) coordinate systems of the left rectangular area and the lower right sector area, (**c**) and the chemical structure of the Kapton membrane used in simulation and experiment.

**Figure 4 polymers-16-00951-f004:**
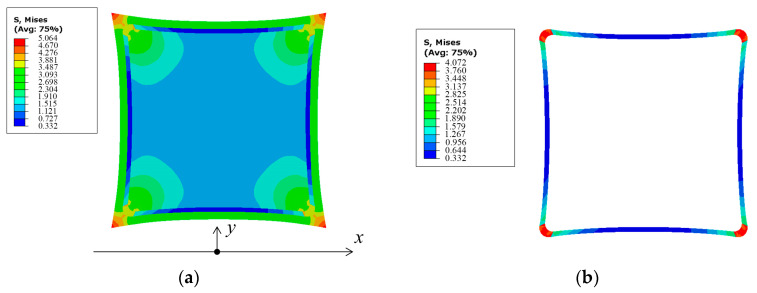
Stress distribution after form-finding of (**a**) membrane and (**b**) shear compliant border.

**Figure 5 polymers-16-00951-f005:**
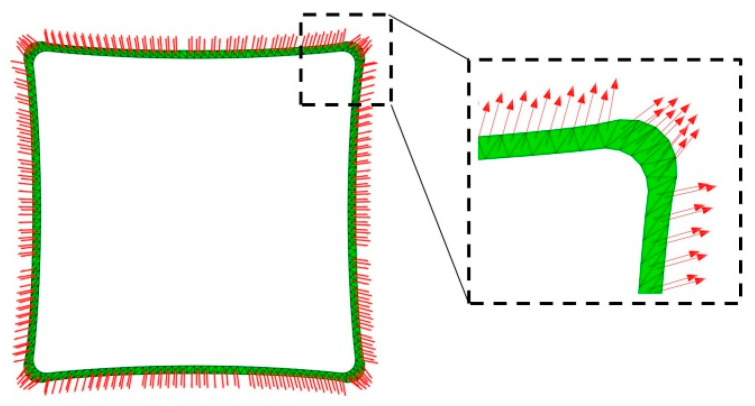
Direction distribution of shear compliant border after form-finding.

**Figure 6 polymers-16-00951-f006:**
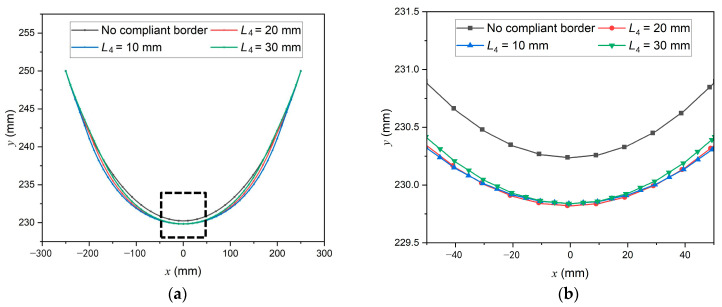
Comparison of (**a**) membrane upper edges and (**b**) central part of the upper edges after form-finding.

**Figure 7 polymers-16-00951-f007:**
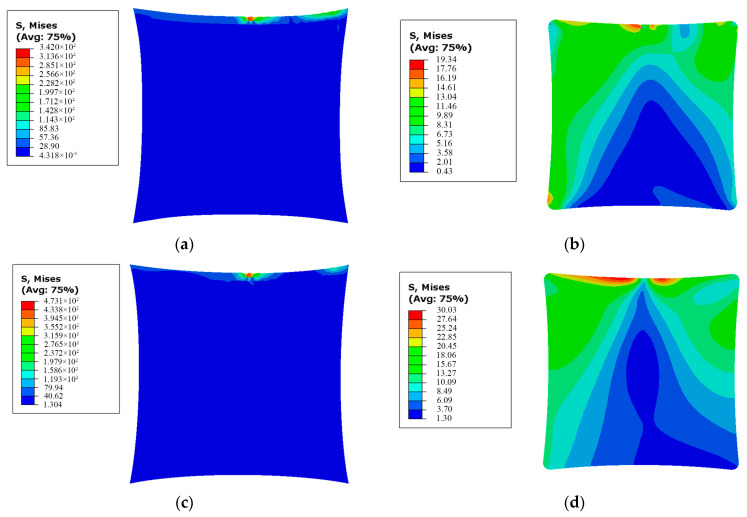
Stress distribution of (**a**) membrane and (**b**) effective area of the membrane structure with shear compliant border and stress distribution of (**c**) membrane and (**d**) effective area of the membrane structure without shear compliant border after applying shear stress.

**Figure 8 polymers-16-00951-f008:**
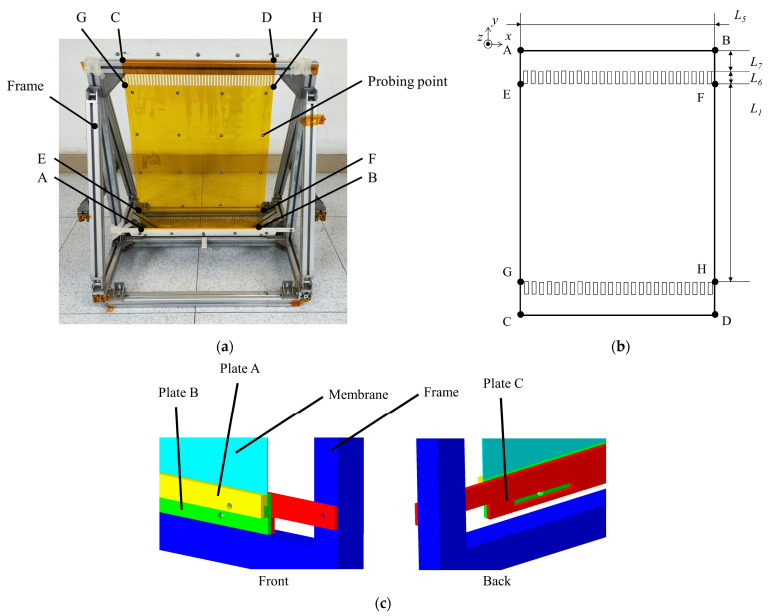
(**a**) Membrane experiment sample and equipment, (**b**) membrane sample and (**c**) fixing components.

**Figure 9 polymers-16-00951-f009:**
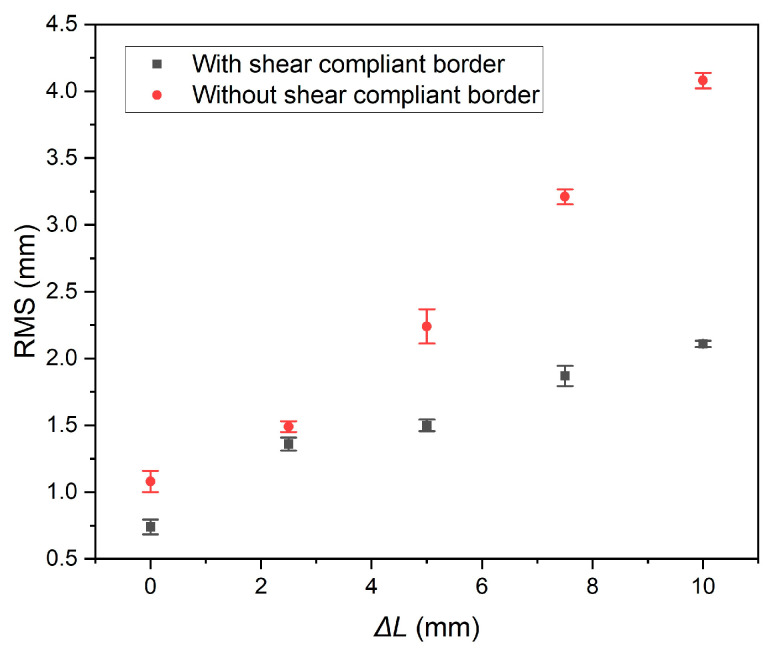
Experiment result of membrane flatness.

**Table 1 polymers-16-00951-t001:** Design parameters of space membrane structure.

Parameter	Value	Parameter	Value
*L* _1_	50 cm	*L* _2_	44 cm
*L* _3_	1 cm	*R* _2_	2 cm
*E* _m_	2.34 GPa	*ν* _m_	0.3
*E* _h_	9.5 MPa	*ν* _h_	0.01
*E* _r_	20 GPa	*ν* _r_	0.3
*t*	25 μm	*r*	0.5 mm
*V* _m_	0.5	*V* _h_	0.5
*L* _4_	2 cm		

**Table 2 polymers-16-00951-t002:** Design parameters of membrane sample.

Parameter	Value	Parameter	Value
*L* _5_	50 cm	*L* _6_	3.5 cm
*L* _7_	5.5 cm	*t*	25 μm
*V* _m_	0.5	*V* _h_	0.5

## Data Availability

The original contributions presented in the study are included in the article, further inquiries can be directed to the corresponding author.

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
