# Peer review of "Configuration Design and Verification of Shear Compliant Border in Space Membrane Structure"

_polymers, 2024, doi:10.3390/polym16070951_

Round 1
Reviewer 1 Report
Comments and Suggestions for Authors
The shear compliant borders are designed, optimized, and verified in terms of configuration by Cao et al which is helps to solve the non-uniformity of stress in space membrane structure. The proposed model act as a reference for stress uniformity design in space membrane structures. I have reviewed the paper thoroughly. The overall presentation of this work is good. Therefore, I accept this work to the Polymers Journal after the following minor comments.
Minor comments are as follows,
- Provide space between number and unit (0.23% to 0.23 %). Check this throughout the manuscript.
- The authors include main objectives and novelty of the present work in the introduction section.
- The authors should cite very recent references related to this work in the introduction section.
- Conclusions needs to be improved further. Give important finding of the results.
- Make sure all abbreviations are written out in full the first time used.
- Avoid abbreviation in the heading and subheading.
- The authors provided many Figures in this work. So, the authors are advised to merge some of the figures. It is better to post the figure numbers less than 10.
Reviewer 2 Report
Comments and Suggestions for Authors
The article: “Configuration Design and Verification of Shear Compliant Border in Space Membrane Structure” raises an important issue related to uniform stress distribution in space membrane structure.
This paper successfully proposed to conduct Shear Compliant Borders (SCB) configuration optimization design and validation (SCB).
The topic is original and relevant for the special issue: “Advanced Polymer Composites in Aerospace Applications”. The shear compliant border configuration design method provides a reference for space membrane structure stress-uniform design.
There is small number of papers published on this issue.
The article may be accepted for publication after minor revision.
The experimental part is quite poor and contains a lot of general information.
More information about the Kapton polymer should be provided, including its manufacturer and chemical structure in the form of a scheme.
Reviewer 3 Report
Comments and Suggestions for Authors
1. You compare your results with Leifer's model [9]. You write: "Using the method from Leifer's paper [9], the calculation of Gzxc takes 9.60×10-4 seconds, resulting in Gzxc = 20.3647 MPa. The difference is ((20.8066-20.3647)/ 20.3647)×100% = 2.17%, and the calculation time is reduced by ((9.60-6.37)/ 9.60)×100% = 33.6%." (lines 167-170). However, without a description of the anisotropy parameters of the Leifer model, this comparison is not quite correct.
2. You write that "The cylindrical coordinate system for the sector region in the bottom-right part is illustrated in the magnified portion of Figure 3..." (lines 190-191). However, the cylindrical coordinate system is three-dimensional. Figure 3 shows a two-dimensional Cartesian coordinate system, not a cylindrical coordinate system. This is misleading to the reader.
3. Before using Hooke's law in the greatly simplified form (15) and (16), the applicability of these simplified equations must be proven. We need to analyze the stress-strain state. In particular, it follows from equation (15) that zz=0. Is this true and why?
4. You write, "If not considering the situation where the deformation region of the SCB is flattened due to excessive shear stress [12], the SCB will not cause wrinkles inside the membrane due to mismatched strains." (lines 212-214). Explain in what situations does this arise? When is it necessary to take this into account? How would this complicate your analysis?
5. You write, "The geometric nonlinearity switch is turned on during the analysis." (lines 219-220). Explain its necessity for the analysis and parameters. The fact that it is turned on is not enough to understand the essence.
6. First you write: "When the maximum displacement of the structure after calculation is less than 1 mm, the result is considered to have converged." (lines 227-228). Next, "The maximum difference in the upper edge between the form-finding results with and without SCB is 1.16 mm..." (lines 249-250). I am alarmed that the figures quoted are the same. Is not the difference of 1.16 mm a methodological error. Judging from the first statement, out of 1.16 mm I can explain 1 mm by convergence error. That leaves 0.16 mm. However, the same order of magnitude will be the rounding error. Thus, I see no objective differences between the curves in Figure 6. An in-depth explanatory analysis of this situation is needed.
7. You write "The number of points is n." (line 278). Specify the number of points you used in your modeling.
8. It is not necessary to connect the experimental points with the lines in Figure 11. Or you need to explain how the points were approximated between measurements.
9. It is not at all clear from Figure 11 how many experiments were performed to obtain the experimental point. You have indicated only the average value. But you need a spread for a correct understanding. Specify the sample mean square deviation for each point in Figure 11. Otherwise, this figure is completely uninformative.
10. References should be added: before formulas (1) and (2) where these expressions are from, before formulas (3) and (4) where these expressions are from, after "According to the Hill composite material model" (line 149), after "According to the Christensen-Lo composite material model" (line 151).
The paper presents a qualitative and good research. However, it needs revision before publication.
Round 2
Reviewer 3 Report
Comments and Suggestions for Authors
I believe that the authors have sufficiently improved the article. It can now be published.